# Facile and Efficient Syntheses of (11*Z*,13*Z*)-Hexadecadienal and Its Derivatives: Key Sex Pheromone and Attractant Components of Notodontidae

**DOI:** 10.3390/molecules24091781

**Published:** 2019-05-08

**Authors:** Fu Liu, Xiangbo Kong, Sufang Zhang, Zhen Zhang

**Affiliations:** Key Laboratory of Forest Protection of National Forestry and Grassland Administration, Research Institute of Forest Ecology, Environment and Protection, Chinese Academy of Forestry, Beijing 100091, China; liufu2006@163.com (F.L.); xbkong@sina.com (X.K.); zhangsf@caf.ac.cn (S.Z.)

**Keywords:** sex pheromone, Notodontide, (*Z*,*Z*)-dienes, conjugated en-yne moieties, total synthesis

## Abstract

Syntheses of (11*Z*,13*Z*)-hexadecadienal (**1**), (11*Z*,13*Z*)-hexadecadienol (**2**), (11*Z*,13*Z*)-hexadecadien-1-yl acetate (**3**), and (Z)-13-hexadecen-11-ynal (**4**) from commercially available starting material 10-bromo-1-decanol are reported. These (*Z*,*Z*)-dienes and conjugated en-yne moieties are common in sex pheromone and attractant components for many Notodontide insect pests. The synthetic scheme, using the C10 + C3 + C3 strategy, was mainly based on three key steps: alkylation of lithium alkyne under a low temperature, *cis*-Wittig olefination of the aldehyde with propylidentriphenylphosphorane, and hydroboration-protonolysis of alkyne. This synthetic route provided (11*Z*,13*Z*)-hexadecadienal (**1**) in a 23.0% total yield via an eight-step sequence, alcohol (**2**) in a 21.9% total yield, acetate (**3**) in a 21.4% total yield, and (Z)-13-hexadecen-11-ynal (**4**) in a 34.7% total yield. This simple strategy provides a new way to achieve syntheses of the key sex pheromones of Notodontide insect pests.

## 1. Introduction

Sex pheromones offer an environmentally-friendly alternative to control insect populations via mating disruption or other strategies in integrated pest management. Notodontidae (Lepidoptera, Noctuoidea) is a family of moths with approximately 3800 known species. Some Notodontids cause noticeable defoliation of their hosts, which causes serious ecological and economic losses [1].

Sex pheromones of ten species of Notodontid have been identified so far [2]. The main component of the sex pheromone of *Oligocentria semirufescens* is (*Z*)-dodec-7-en-1-ol, which is also attractive to the male moth of *Schizura semirufescens* [3]. The sex pheromone of *Thaumetopoea bonjeani* is a mixture of (11*Z*,13*Z*)-hexadecadienal (**1**) and (11*Z*,13*Z*)-hexadecadienol (**2**), and the ratio is 4:1 [4]. However, for another sympatric species, such as *T. pityocampa*, *T. wilkinsoni*, and *T. processionea*, (*Z*)-13-hexadecen-11-yn-1-yl acetate is the key sex pheromone candidate compound, and possesses a conjugated en-yne moiety with a *Z* configuration, which is a new structure in the insect pheromone field [5,6,7]. In comparison, (*Z*)-13-hexadecen-11-ynal (**4**) is the sex pheromone component of *Heterocampa guttivitta* [8]. (11*Z*,13*Z*)-hexadecadienal (**1**) and (11*Z*,13*Z*)-hexadecadien-1-yl acetate (**3**) are the active sex pheromone ingredients of *N. dromedaries* and *N. torva*, respectively [9,10]. Trace-chemical reaction and coupled gas chromatography-mass spectrometry (GC-MS) analyses show that the active component of *Clostera anastomosis* is tetradecadienal [11].

Accordingly, most female-produced sex pheromones of Notodontid are normally complex mixtures of straight chain acetates, aldehydes, and alcohols, with 16 carbon atoms. This group of pheromones belongs to Type I, according to Ando’s classification [12].

The structures of (11*Z*,13*Z*)-hexadecadienal (**1**), (11*Z*,13*Z*)-hexadecadienol (**2**), and (11*Z*,13*Z*)-hexadecadien-1-yl acetate (**3**) contain a conjugated (*Z*,*Z*)-diene moiety. The construction of the (*Z*,*Z*)-conjugated diene system has been achieved in the past using a series of schemes: (1) the Wittig reaction: addition reaction of aldehyde with a *Z* type-unsaturated enol [13]; (2) Cadiot–Chodkiewicz coupling coupled with hydroboration-protonolysis of 1,3-diyne [14]; (3) diimide reduction of acetylene and palladium-catalyzed coupling [15]; and (4) catalytic coupling of metal catalysts and hydroboration-protonolysis [16,17,18,19,20,21,22]. 

As for (*Z*)-13-hexadecen-11-ynal (**4**), it possesses a conjugated en-yne moiety of *Z* configuration. The presently reported synthetic scheme for a conjugated en-yne moiety mainly includes: (1) the Wittig condensation of the propargylic aldehyde with propylidentriphenylphosphorane [23] and (2) cross-coupling of vinyl copper lithium with iodoalkynes [24].

Highly selective methods have to be employed for constructing the (*Z*,*Z*)-conjugated diene system and conjugated en-yne moiety of a *Z* configuration, which exist in the structure of (11*Z*,13*Z*)-hexadecadienal (**1**) and (*Z*)-13-hexadecen-11-ynal (**4**), respectively. Furthermore, both (11*Z*,13*Z*)-hexadecadienal (**1**) and (*Z*)-13-hexadecen-11-ynal (**4**) possess the labile terminal formyl group, therefore the timing of placing the terminal formyl group is also important, owing to the fact that oxidation in the final step may isomerize the (*Z*,*Z*)-conjugated diene system [22]. The formyl group is introduced into its intermediate as acetal derivatives in an earlier stage of synthesis, which has been shown to avoid isomerization of the (*Z*,*Z*)-conjugated diene system from oxidation [17,18,19,20].

Consequently, we have successfully established a C10 + C3 + C3 synthetic strategy to efficiently obtain (11*Z*,13*Z*)-hexadecadienal (**1**), (11*Z*,13*Z*)-hexadecadienol (**2**), (11*Z*,13*Z*)-hexadecadien-1-yl acetate (**3**), and (*Z*)-13-hexadecen-11-ynal (**4**) together, using 1,1-diethoxy-10-iododecane (**5**), 13,13-diethoxytridec-2-ynal (**6**), and (*Z*)-16,16-diethoxyhexadec-3-en-5-yne (**7**) as the key building blocks, employing the alkylation of lithium alkyne, electrolytic manganese dioxide oxidation, *cis*-Wittig olefination reaction, and hydroboration-protonolysis as the key reactions.

## 2. Results and Discussion 

As Scheme 1 shows, the retrosynthesis was based on the alkylation of lithium alkyne, a *cis*-Wittig olefination reaction, and hydroboration-protonolysis, namely a C10 + C3 + C3 strategy. The two C3 synthons were easily obtained, and the C10 subunit was a protected iodoalkane (**5**), which could be prepared from 10-bromo-1-decanol.

Scheme 1 shows the synthetic plan for (11*Z*,13*Z*)-hexadecadienal (**1**) and its derivatives. *cis*-Wittig olefination of 13,13-diethoxytridec-2-ynal (**6**) with propylidentriphenylphosphorane will give the required carbon-skeleton as (*Z*)-16,16-diethoxyhexadec-3-en-5-yne (**7**), whose deprotection will give (*Z*)-13-hexadecen-11-ynal (**4**), and reduction and deprotection will give (11*Z*,13*Z*)-hexadecadienal **1** and its derivatives (**2** and **3**). The two C3 synthons are commercially available, while the C10 subunit (**5**) can be prepared from 10-bromo-1-decanol.

The facile and efficient synthesis of (11*Z*,13*Z*)-hexadecadienal (**1**), (11*Z*,13*Z*)-hexadecadienol (**2**), (11*Z*,13*Z*)-hexadecadienyl acetate (**3**), and (*Z*)-13-hexadecen-11-ynal (**4**) is summarized in Scheme 2.

Commercially available 10-bromo-1-decanol was chosen as the starting material. 1,1-diethoxy-10-iododecane (**5**) could be obtained in three steps according to Furber’s method [20]. First, 10-bromo-1-decanol was oxidized using pyrindium chlorochromate in CH_2_Cl_2_ at room temperature for 3 h. The clean formation of aldehyde was treated with triethylorthoformate and *p*-toluenesulfonic acid in anhydrous ethanol. Following this, the crude product was refluxed with NaI in anhydrous acetone until the end of the halogen exchange reaction (4 h). The yield of **5** was 78% based on 10-bromo-1-decanol.

Alkynes alkylation is widely used in the synthesis of terminal alkynes and alkynols [22,25,26,27], especially sex pheromones of insect pests [28,29,30]. At the condition of −40 °C under argon, alkylation of lithium propargyl alcohol (C3 synthon) with 1,1-diethoxy-10-iododecane (**5**) in tetrahydrofuran/hexamethyl phosphoryl triamide furnished the acetylenic compound. 

It is well-known that activated manganese dioxide is a useful reagent for the oxidation of unsaturated alcohols to corresponding aldehyde. However, its quality varies widely, the preparation is tedious, and the commercial reagent is expensive. Electrolytic manganese dioxide (EMD) is less expensive and does not require purification [31]. The application of EMD would provide a good option for sex pheromones syntheses. Treatment of the acetylenic compound with excess electrolytic manganese dioxide (30 eq) in hexane at room temperature for 2 h, led to the clean formation of the expected alkynal **6** in a 67% yield based on **5**. 

Wittig reactions are most commonly used to couple aldehydes to singly-substituted phosphine ylides. Unstabilized ylides, mainly via the erythro betaine intermediate, lead to the *Z*-alkene product [29,32,33]. The carbonyl olefination of the alkynal **6** with phosphorus ylides is a general method for the preparation of *Z*-enyne **7**. To ensure the *cis* selectivity of Wittig olefination, potassium bis(trimethylsilyl)amide was chosen as the base. The ylide, prepared from *n*-propyl triphenylphosphonium bromide via a reaction with potassium bis(trimethylsilyl)amide as the base, in a stoichiometric ratio of reagents, reacted with **6** in THF at −70 °C to give (*Z*)-16,16-diethoxyhexadec-3-en-5-yne (**7**). Pure product **7** was isolated from the reaction mixture using column chromatography in a good yield (70%). 

*Z*-selective reduction of the triple bond was a challenge in the synthesis of the sex pheromones. Bercot et al. employed selective catalytic hydrogenation, generally using a Lindlar catalyst [34], but the pretreatment of calcium carbonate carriers is rarely reported. Khrimian et al. first employed zinc activated with copper and silver in *cis* reduction of the conjugated trienynes in pheromone synthesis [35]. Unfortunately, pretreatment of the active Zn reagent is complicated and time-consuming. Hungerford and Kitching applied titanium (II) to finish the triple reduction [36]. However, the reduction gave additional products generated via the 1,4-reduction of the en-yne. As previously reported, 3–4 equivalents of alkylborane in THF are necessary to effectively complete hydroboration of the triple bond [14,22]. The reaction system was treated with acetic acid to achieve protonolysis. Oxidation of the resulting dicyclohexylborinate was achieved via the addition of aqueous sodium hydroxide followed by the dropwise addition of hydrogen peroxide. The crude product contained **1** as well, which was liberated in the course of protonolysis with acetic acid. In addition, without purification, the mixture was dissolved in THF and treated with aqueous oxalic acid to deprotect the acetal moiety and give **1** in a 63% yield based on **7**.

The compound **1** was cleanly converted into the corresponding alcohol **2** in a 95% isolated yield by reduction with LiAlH_4_ in THF under argon.

(11*Z*,13*Z*)-hexadecadienyl acetate (**3**) was easily prepared from (11*Z*,13*Z*)-hexadecadienol (**2**), acetic anhydride, and pyridine in CH_2_Cl_2_ in a yield 98%.

(*Z*)-16,16-diethoxyhexadec-3-en-5-yne (**7**) was dissolved in THF and treated with aqueous oxalic acid to deprotect the acetal moiety and give (*Z*)-13-hexadecen-11-ynal (**4**) in a 95% yield.

## 3. Materials and Methods 

### 3.1. Chemistry

#### 3.1.1. General Method

All commercially available reagents were used without further purification. THF was distilled from sodium. CH_2_Cl_2_ was distilled from CaH_2_. Column chromatography was performed on silica gel (≈200–400 mesh). ^1^H-NMR (500 MHz) and ^13^C-NMR (125 MH_Z_) spectra were recorded on a Bruker NMR spectrometer (Bruker, Fällanden, Switzerland). The component analysis was carried out using an Agilent gas chromatograph coupled with a mass spectrometry system (TRACE GC 2000). The GC was equipped with a polar HP-5MS column (30 m × 0.25 mm × 0.25 μm, Agilent Technologies, Wilmington, DE, USA) and included an injector temperature set to 230 °C. The oven temperature for the HP-5MS GC column was initially programmed at 60 °C for one minute and then subsequently increased to 280 °C at 8 °C per minute. Helium was used as the carrier gas. The GC data for compounds **1**, **2**, **3**, **4**, and NMR spectra for all synthetic compounds were list in the Appendix A.

#### 3.1.2. General Procedure for the Synthesis of Compounds

*1,1-Diethoxy-10-iododecane (***5***)*: 10-Bromodecanol (50 g, 210 mmol) was oxidized using pyridinium chlorochromate (90.5 g, 420 mmol) in CH_2_Cl_2_ (800 mL) at room temperature for 3 h. The organic layer was filtered, and the residual was washed with petroleum. Removal of the solvent from the combined organic layers in vacuo gave a dark oil. The dark-colored residue was chromatographed over SiO_2_, and elution with hexane/EtOAc (30:1, *v/v*) gave crude 10-bromodecanal.

Triethyl orthoformate (44.5g, 300 mmol) and *p*-sulphonic acid monohydrate (0.57 g, 3 mmol) were added to a stirred and ice-cooled solution of the crude 10-bromodecanal in anhydrous ethanol (300 mL). After the exothermic reaction had subsided, the mixture was left at 0 °C overnight. Water was then added, and the mixture was made basic by adding K_2_CO_3_ solution. The mixture was extracted with diethyl ether, and then washed with brine and dried over MgSO_4_. The solvent was removed under reduced pressure and the product was chromatographed on silica (hexane/EtOAc (25:1, *v/v*)), which gave crude 1,l-diethoxy-10-bromodecanal.

This product was converted into the title iodide by being stirred for 4 h with sodium iodide (90 g, 600 mmol) in dry acetone (500 mL) under reflux. The solvent was removed under reduced pressure, the mixture was diluted with water (200 mL), and the product was extracted with petroleum. The extracts were washed with water, 1% Na_2_S_2_O_3_ solution, and brine; dried over Na_2_SO_4_; and concentrated under reduced pressure. The resulting residue was chromatographed over SiO_2_. Elution with hexane/EtOAc (25:1, *v/v*) was conducted to yield 1,l-diethoxy-10-iododecan (**5**) as an oil (58.8 g, 78% yield based on 10-bromodecanol); ^1^H-NMR (500 MHz, CDCl_3_) δ 1.21 (6H, t, *J* = 7.0 Hz), 1.29 (12 H, m), 1.60 (2 H, m), 1.82 (2H, m), 3.19 (2H, t, *J* = 7.0 Hz), 3.49 (2H, m), 3.64 (2H, m), 4.48 (1H, t, *J* = 6.0 Hz); ^13^C-NMR (125 MHz, CDCl_3_) δ 102.9, 60.8, 60.8, 33.6, 33.5, 30.5, 29.4, 29.4, 29.3, 28.5, 24.7, 13.4, 15.4, 7.3.

*13,13-Diethoxytridec-2-ynal (***6***)*: *n*-Butyllithium (2.5 M in hexane) (160 mL, 400 mol) was added slowly to a solution of propargyl alcohol (11.2 g, 200 mol) in HMPT/THF (1:1, *v/v*, 800 mL) at −40 °C under argon, and the solution was stirred for 30 min. A solution of 1,l-diethoxy-10-iododecan (**5**) (35.6 g, 100 mmol) in HMPT/THF (1:1, *v/v*, 50 mL) was then added over 20 min. After being stirred overnight at the same temperature (−20 °C), the reaction mixture was quenched with water and extracted with ethyl acetate. The organic layer was washed with water and dried with Na_2_SO_4_. Evaporation left the crude product. 

An excess of electrolytic manganese dioxide (52.1 g, 600 mmol) was added to a solution of the crude product (17.0 g) in dry hexane (300 mL). The mixture was stirred at room temperature for 4 h, and the residues containing manganese were filtered off. The yield of the protected alkenal **6** after chromatography was 67% (15.7 g) in two steps. ^1^H-NMR (500 MHz, CDCl_3_) δ 1.19–1.26 (6H, m), 1.29–1.30 (12H, m), 1.40 (2H, m), 1.58–1.63 (2H, m), 2.40–2.43 (2H, m), 3.46–3.52 (2H, m), 3.61–3.67 (2H, m), 4.48 (1H, t, *J* = 6.0 Hz), 9.77 (1H, t, *J* = 1.5 Hz); ^13^C-NMR (125 MHz, CDCl_3_) δ 177.1, 102.8, 91.4, 75.6, 60.5, 60.5, 33.5, 29.3, 29.3, 29.2, 29.0, 28.8, 28.7, 24.6, 18.5, 15.2, 15.2.

*(Z)-16,16-diethoxyhexadec-3-en-5-yne (***7***)*: *n*-propyl triphenylphosphonium bromide (23.1 g, 60 mmol) in THF (150 mL) was added to a solution of potassium bis(trimethylsilyl)amide (0.5 M in toluene, 140 mL, 70 mmol). After refluxing for 1 h, the reaction mixture was cooled to −70 °C, and a solution of the aldehyde **6** (14.1 g, 50 mmol) in THF (20 mL) was added dropwise. The mixture was then stirred for 3 h, before it was poured into aqueous NH_4_Cl (10%, 30 mL). The organic phase was separated and the aqueous phase was extracted with hexane. The combined organic phases were dried with Na_2_SO_4_. After evaporation, the crude product was subjected to medium-pressure column chromatography (30:1, *v/v*), yielding 70% (10.8 g). ^1^H-NMR (500 MHz, CDCl_3_) δ 1.01 (3H, t, *J* = 7.5 Hz), 1.20 (6H, t, *J* = 7.0 Hz), 1.29 (12H, m), 1.51–1.55 (2H, m), 1.59–1.63 (2H, m), 2.28–2.35 (4H, m), 3.49 (2H, m), 3.64 (2H, m), 4.48 (1H, t, *J* = 6.0 Hz), 5.40 (1H, m), 5.80 (1H, m); ^13^C-NMR (125 MHz, CDCl_3_) δ 144.0, 108.7, 102.9, 94.5, 77.2, 60.8, 60.8, 33.6, 29.5, 29.5, 29.4, 29.1, 28.9, 28.8, 24.7, 23.4, 19.5, 15.3, 15.3, 13.4.

*(11Z,13Z)-hexadecadienal (***1***)*: A solution of compound **7** (6.2 g, 20 mmol) in THF (10 mL) was added dropwise to the above dicyclohexylborane solution (42 mmol) at −20 °C. The suspension was stirred at approximately −15 °C for 2 h and then allowed to reach room temperature. After 2 h of stirring at room temperature, the precipitate of dicyclohexylborane had disappeared. Glacial acetic acid (5 mL) was then added to the mixture, which was stirred for 2 h at 50 °C. Oxidation of the resulting dicyclohexylborinate was achieved by the addition of sodium hydroxide (6 M, 6 mL) followed by the dropwise addition of hydrogen peroxide (35%, 7 mL). The mixture was stirred for an additional 30 min and was then poured into ice-water (15 mL), extracted with hexane, and dried (MgSO_4_). After evaporation, a solution of the crude product in tetrahydrofuran (30 mL) was added to a solution of oxalic acid dihydrate (3.0 g) in water (30 mL). The mixture was stirred and heated for 40 min at 60 °C under argon. Then, the mixture was extracted with hexane. The organic solution was washed with water, a sodium hydrogen carbonate solution, and brine; dried (Na_2_SO_4_); and concentrated in vacuo. The residue was chromatographed over SiO_2_ with hexane/EtOAc (20:1, *v/v*), which gave **1** (3.0 g, 63%) as a colorless oil. ^1^H-NMR (500 MHz, CDCl_3_) δ 1.00 (3H, t, *J* = 7.5 Hz), 1.28–1.38 (12H, m), 1.60–1.66 (2H, m), 2.15–2.44 (4H, m), 2.42 (2H, td, *J* = 7.5, 1.5 Hz), 5.42–5.47 (2H, m), 6.19–6.28 (2H, m), 9.77 (1H, t, *J* = 2.0 Hz); ^13^C-NMR (125 MHz, CDCl_3_) δ 202.9, 133.6, 132.1, 123.4, 123.0, 43.9, 29.6, 29.4, 29.3, 29.3, 29.2, 29.1, 27.4, 22.0, 20.8, 14.2. GC-MS: *t*_R:_ 21.18 min; MS of **1** (70 eV, EI): 236.

*(11Z,13Z)-hexadecadienol (***2***)*: A 100 mL dried flask was charged with freshly prepared THF (30 mL) under argon and cooled to 0 °C, while LiAlH_4_ (760 mg, 20 mmol) was added in portions. A solution of 2.36 g (10 mmol) of **1** in 10 mL THF was added to the flask using a syringe. The resulting mixture was stirred for 30 min at 0 °C and then warmed to room temperature. The reaction was monitored using GC until the peak of **1** disappeared. The reduction mixture was cooled with an ice bath and treated via the successive dropwise addition of water and 15% sodium hydroxide solution. The dry granular precipitate was removed via filtration, the filtrate was dried over Na_2_SO_4_, and the solvent was evaporated. The crude material was purified using flash chromatography on silica gel using hexane-ethyl acetate (15:1, *v/v*) as an eluent to provide 2.26 g (9.5 mmol, 95% yield) of **2** as a colorless liquid. ^1^H-NMR (500 MHz, CDCl_3_) δ 1.01 (3H, t, *J* = 7.5 Hz), 1.29-1.40 (16H, m), 1.54–1.59 (2H, m), 2.16–2.21 (2H, m), 3.64 (2H, t, *J* = 7.5 Hz), 5.42–5.48 (2H, m), 6.20–6.28 (2H, m); ^13^C-NMR (125 MHz, CDCl_3_) δ 133.6, 132.1, 123.4, 123.0, 63.1, 32.8, 29.6, 29.6, 29.5, 29.5, 29.4, 29.3, 27.5, 25.7, 20.8, 14.2. GC-MS: *t*_R:_ 20.68 min; MS of **2** (70 eV, EI): 238.

*(11Z,13Z)-hexadecadienyl acetate (***3***)*: A total of 0.47 g (6 mmol) of pyridine was added to a solution of 1.19 g (5 mmol) of **2** in 10 mL of CH_2_Cl_2_ at 0 °C, followed by 0.61 g (6 mmol) of acetic anhydride. The mixture was stirred for 4 h and washed with water. The organic was then concentrated to remove the solvent, and the crude was purified using flash chromatography on silica gel using hexane-ethyl acetate (30:1, *v/v*) as an eluent to provide 1.37 g (4.9 mmol, 98%) of **3** as a colorless liquid. ^1^H-NMR (500 MHz, CDCl_3_) δ 1.01 (3H, m), 1.31 (16H, m), 1.62 (2H, m), 2.04 (3H, s), 2.16–2.19 (2H, m), 4.05 (2H, t, *J* = 6.5 Hz), 5.42–5.47 (2H, m), 6.21–6.28 (2H, m); ^13^C-NMR (125 MHz, CDCl_3_) δ 171.3, 133.6, 132.1, 123.4, 123.0, 64.7, 32.9, 29.7, 29.6, 29.5, 29.5, 29.3, 29.3, 28.6, 27.5, 25.9, 21.0, 14.2. GC-MS: *t*_R:_ 21.97 min; MS of **3** (70 eV, EI): 280.

*(Z)-13-hexadecen-11-ynal (***4***)*: A solution of **7** (3.1 g, 15 mmol) in tetrahydrofuran (20 mL) was added to a solution of oxalic acid dihydrate (3.0 g) in water (10 mL). The mixture was stirred and heated for 40 min at 60 °C under argon. Then, the mixture was extracted with hexane. The organic solution was washed with water, a sodium hydrogen carbonate solution, and brine; dried (Na_2_SO_4_); and concentrated in vacuo. The residue was chromatographed over SiO_2_ with hexane/EtOAc (30:1, *v/v*), which gave **4** (2.24 g, 95%) as a colorless oil. ^1^H-NMR (500 MHz, CDCl_3_) δ 1.00 (3H, t, *J* = 7.5 Hz), 1.28–1.38 (12H, m), 1.60–1.66 (2H, m), 2.15–2.44 (4H, m), 2.42 (2H, td, *J* = 7.5, 1.5 Hz), 5.42–5.47 (1H, m), 6.19–6.28 (1H, m), 9.77 (1H, t, *J* = 2.0 Hz); ^13^C-NMR (125 MHz, CDCl_3_) δ 202.9, 133.6, 132.1, 123.4, 123.0, 43.9, 29.6, 29.4, 29.3, 29.3, 29.2, 29.1, 27.4, 22.0, 20.8, 14.2. GC-MS: *t*_R:_ 21.83 min; MS of **4** (70 eV, EI): 234.

## 4. Conclusions

Based on a C10 + C3 + C3 strategy, facile and efficient syntheses of (11*Z*,13*Z*)-hexadecadienal (**1**)**,** alcohol (**2**)**,** corresponding acetate (**3**)**,** and (*Z*)-13-hexadecen-11-ynal (**4**), which are key sex pheromone and attractant components of Notodontidae, were achieved. The key steps were accomplished by the alkylation of lithium alkyne, a *cis*-Wittig olefination reaction, and hydroboration-protonolysis. In addition, (11*Z*,13*Z*)-hexadecadienal (**1**) was also identified as the sex pheromone component of navel orangeworm, *Amyelois transitella* (Pyralidae) [13], which has become a key pest of tree nuts in California [37,38]. Additionally, for the meal moth *Pyralis farinalis*, a blend of (11*Z*,13*Z*)-hexadecadienal (**1**) and (3*Z*,6*Z*,9*Z*,12*Z*,15*Z*)-tricosapentaene acts as the attractant, while (11*Z*,13*Z*)-hexadecadien-1-yl acetate (**3**) is a behavioral antagonist [39,40]. This simple, convenient, and efficient synthetic route will be greatly helpful for the further practical testing and the use of pheromones as benign environmental tools for the pest control of Notodontidae and Pyralidae.

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
