# Peer review of "Facile and Efficient Syntheses of (11Z,13Z)-Hexadecadienal and Its Derivatives: Key Sex Pheromone and Attractant Components of Notodontidae"

_molecules, 2019, doi:10.3390/molecules24091781_

Round 1
Reviewer 1 Report
Fu Liu, et al; have nicely described a simple and adaptable strategy to synthesize four of the important analogues of sex pheromones from Notodonfidae family.
Strategy for synthesis was well drafted and executed.
Manuscript was supported with adequate analytical data (NMR, mass etc)
I recommend publication of the manuscript in Molecules journal after minor revisions.
1. Development of routes/synthesis of pheromone compounds is very important area of research which is an alternative and effective way to reduce the usage of pesticides for crops.
2. Authors used electrolytic MnO2 at oxidation of unsaturated alcohols an effective way for the synthesis of all the pheromones they synthesized here from simple starting materials.
3. Here, a C10+C3+C3 strategy works well across all the pheromone molecules they synthesized.
4. Reported yields for each step further strengthens the C10+C3+C3 strategy that author adopts.
5. However, I recommend report the HRMS / elemental analysis data for each step for each fragment in the strategy.
6. I recommend mention the manuscript title and authors name in the supplementary information as well.
7. I recommend publish the manuscript in Molecules journal.
Author Response
Thank you very much for your advice regarding to our manuscript. We have tried best to revise our manuscript according to the comments. some questions have been clarified as follows:
Point 1: Development of routes/synthesis of pheromone compounds is very important area of research which is an alternative and effective way to reduce the usage of pesticides for crops.
Response 1: Thanks for your appreciation..
Point 2: Response 2: Thanks for your comment.
Point 3: Here, a C10+C3+C3 strategy works well across all the pheromone molecules they synthesized.
Response 3: Thanks for your comment.
Point 4: Reported yields for each step further strengthens the C10+C3+C3 strategy that author adopts.
Response 4: Thanks for your appreciation.
Point 5: However, I recommend report the HRMS / elemental analysis data for each step for each fragment in the strategy.
Response 5: Thanks for your question. We have added the GC-MS data (Compound 1~4) in our revised version and the supplementary information.
Point 6: I recommend mention the manuscript title and authors name in the supplementary information as well.
Response 6: Thanks for your suggestion. We have added the manuscript title and authors name in the supplementary information as well.
Point 7: I recommend publish the manuscript in Molecules journal.
Response 7: Thanks for your appreciation.

Reviewer 2 Report
Type of manuscript: Article
Title: Facile and Efficient Syntheses of (11Z,13Z)-Hexadecadienal and its
Derivatives: Key Sex Pheromone and Attractant Components of Notodonfidae
Journal: Molecules
Review: This is a well written article using some standard organic syntheses to construct conjugated enes and ene-ynes in the chemical ecology field. The paper is generally well written and the data supports their conclusions. Accept with minor revsions.
Generalcomments:
1] The family name is often misspelled throughout the manuscript.. Notodontidae….t not f.
2] line 40…tetradecadienyl aldehyde??? Why not tetradecadienal??
3] lines 60 and 62….”damage” is not an appropriate word. In this context. What do you mean here??. Oxidise, ……..???
4] The synthetic schemes are well layed out and very clear.
5] line 149….you mean “residue” ??
6] line 151. ..monohydrate. Spelling.
7] line 210..freshly prepared.

Author Response
Point 1: The family name is often misspelled throughout the manuscript. Notodontidae….t not f.
Response 1: Thank you for your positive advice. We have revised “Notodonfidae” to “Notodontidae” in our manuscript.
Point 2: line 40…tetradecadienyl aldehyde??? Why not tetradecadienal??
Response 2: Thanks for your positive advice. We have changed “tetradecadienyl aldehyde” to “tetradecadienal”.
Point 3: lines 60 and 62….”damage” is not an appropriate word. In this context. What do you mean here??. Oxidise, ……..???
Response 3: Thanks for your suggestion. We have deleted “damage” in our manuscript. The (Z,Z)-conjugated diene is unstable, the oxidation of alcohol to the formyl group may isomerize the (Z,Z)-conjugated diene system. Furthermore, the timing of placing the terminal formyl group is also important.
Point 4: The synthetic schemes are well layed out and very clear.
Response 4: Thanks for your appreciation.
Point 5: line 149….you mean “residue” ??
Response 5: Thanks for your advice. We have changed “reside” to “residue” in our manuscript.
Point 6: line 151. ..monohydrate. Spelling.
Response 6: Thanks for your addvice. We are really sorry that we didn’t spell it correctly. We have revised it in our manuscript.
Point 7: line 210..freshly prepared.
Response 7: Thanks for your advice. We have modified it in our manuscript.

Reviewer 3 Report
Zhen Zhang et al described a facile and efficent syntheses of (11Z,13Z)-2 Hexadecadienal. this is a component for a insect pheromone. As we all know, the synthesis for special cis alkene is hard normally. The authors had used different strategies to built the cis alkene, and it was pure enough to aplly. Based on the above points, I am gald to recomment to publish this paperAuthor Response
Point : Zhen Zhang et al described a facile and efficent syntheses of (11Z,13Z)-2 Hexadecadienal. this is a component for a insect pheromone. As we all know, the synthesis for special cis alkene is hard normally. The authors had used different strategies to built the cis alkene, and it was pure enough to aplly. Based on the above points, I am gald to recomment to publish this paper
Response 1: Thanks for your advice and appreciation.
